# Exogenous Xyloglucan Oligosaccharides Alleviate Cadmium Toxicity in *Boehmeria nivea* by Increasing the Cadmium Fixation Capacity of Cell Walls

Yushen Ma [1], Hongdong Jie [1], Long Zhao [1], Ying Zhang [1], Pengliang He [1], Xueying Lv [1], Xiaochun Liu [1], Yan Xu [1] and Yucheng Jie [1,2,*]

[1] College of Agronomy, Hunan Agricultural University, Changsha 410128, China; mys9204@stu.hunau.edu.cn (Y.M.); jhd20210218@stu.hunau.edu.cn (H.J.); azlhh@stu.hunau.edu.cn (L.Z.); zhangying@hnlky.cn (Y.Z.); hpl888@stu.hunau.edu.cn (P.H.); sx20210341@stu.hunau.edu.cn (X.L.); wzx1987@stu.hunau.edu.cn (X.L.); 2455701019@stu.hunan.edu.cn (Y.X.)

[2] Hunan Provincial Key Laboratory of Crop Germplasm Innovation and Utilization, Changsha 410128, China

\* Correspondence: fbfcjyc@hunau.edu.cn

**Abstract:** Xyloglucan is an important component of hemicellulose, and xyloglucan oligosaccharides (Xh), which are metabolized by xyloglucan, play an important role in plant growth and development. However, the regulatory effects of the external application of Xh under cadmium (Cd) stress have not been determined. In this study, we evaluated the mechanism by which Xh contributes to resistance to Cd stress in ramie, a candidate plant species for toxic ion removal. The external application of Xh effectively attenuated the effects of Cd on ramie growth and photosynthetic pigments. Cd stress can also inhibit the activity of antioxidant enzymes such as superoxide dismutase (SOD), catalase (CAT), peroxidase (POD), and ascorbate peroxidase (APX), resulting in a significant increase in the extent of membrane lipid peroxidation. After the external application of Xh, antioxidant enzyme activity was up-regulated, and damage to membranes in plants was reduced. In addition, the external application of Xh increased Cd retention in roots, thereby significantly decreasing Cd content in shoots. The external application of Xh also regulated the subcellular distribution of Cd and increased the Cd content of the cell wall. In particular, a root cell wall analysis revealed that Cd+Xh treatment significantly increased the hemicellulose content in the cell wall and the amount of Cd retained. In summary, the external application of Xh alleviates Cd toxicity in ramie by increasing the hemicellulose content and the Cd fixation ability of the cell wall and by reducing membrane lipid peroxidation via antioxidant enzymes.

**Keywords:** xyloglucan oligosaccharides; ramie; subcellular distribution; hemicellulose; cell wall; cadmium

## 1. Introduction

Cadmium (Cd), a bivalent toxic metal, is absorbed by roots and accumulates in plants [1]. Excessive Cd will cause chlorosis of leaves, decreased photosynthesis, shortened roots, weakened transpiration, and even plant death in severe cases [2]. In addition, Cd in plants can eventually accumulate in humans via the food chain, threatening human health [3]. In China, the area of cultivated land polluted by Cd exceeds $1.3 \times 10^9 \ m^2$, mainly in the south [4]. Southern China is the main rice-producing area, and rice containing Cd has caused widespread food safety concerns in the country [5]. Cd pollution has become an urgent environmental issue.

Ramie (*Boehmeria nivea*) is a kind of phloem fiber crop with strong Cd enrichment ability, high biomass, and a rapid growth rate. Ramie is a promising plant material for the remediation of Cd-contaminated soil [6]. To thrive in soil contaminated with Cd, plants have evolved a diverse array of defense mechanisms, including Cd efflux, cell wall fixation, the synthesis of plant chelates, and vacuolar compartmentalization [7]. The cell

wall binds metal ions and affects the distribution of metal ions in cells [8]. Studies have reported the significant role of the cell wall in the ramie response to Cd stress; the cell wall contributes to 48.2–61.9% of Cd accumulation in ramie [9]. The chemical components in the cell wall are altered under Cd stress [10]. Our previous studies have shown that cell wall polysaccharides play a decisive role in Cd accumulation, in which Cd enrichment by ramie hemicellulose polysaccharides accounts for more than 56.46% of the total Cd accumulation in the cell wall; at the same time, an increase in hemicellulose content could enhance the ability to accumulate Cd in the cell wall [11]. Therefore, increasing the hemicellulose content by exogenous application or bioengineering is a good strategy to enhance Cd accumulation in the ramie cell wall.

Xyloglucan is an important component of hemicellulose and is widely found in higher plants [12]. It accounts for approximately 20–25% of the cell wall in dicotyledons and is the most abundant hemicellulose component, but is less abundant in grasses, accounting for about 2–5% [13]. Xyloglucan can change or maintain the shape of plant cells [12]. Xyloglucan oligosaccharides (Xh) make up xyloglucan and play a role in the regulation of plant growth and development [14,15]. The external application of Xh can increase plant resistance to abiotic stress, especially waterlogging stress and cold injury [16]. Some studies have shown that Xh can act as a signaling molecule to regulate plant growth and development, cell elongation, and plant defense responses [17]. Recent studies have also shown that xyloglucan and its oligosaccharides can trigger MAPK activation and immune gene expression to trigger an immune response [18]. However, little is known about the mechanism by which Xh contributes to plant stress resistance, especially resistance to Cd toxicity. Our preliminary experiment revealed that under Cd stress, plant growth and root length of ramie seedlings treated with Xh were larger than those of seedlings without Xh treatment [11]. To explain these results, we hypothesized that the exogenous application of Xh pr enhances the interaction between ramie cell wall polysaccharides and Cd, consequently augmenting the species' tolerance towards Cd. In this study, to evaluate this hypothesis, the ramie genotype 'Zhongzhu No. 1' was used for analyses of the effect of Xh on plant physiology and biochemistry, Cd subcellular and cell wall polysaccharides, and Cd binding to each cell wall component. Our results provide new insights into the function of cell wall polysaccharides in the detoxification of Cd in ramie.

## 2. Materials and Methods

### 2.1. Plant Growth and Cd Treatment

The Ramie genotype 'Zhongzhu No. 1' was obtained from Hunan Agricultural University. Ramie cuttings from pants with a similar height and stem diameter were placed in flowerpots filled with perlite, with 3 plants per pot. Prior to planting, the cuttings were pre-cultured in a 1/2 Hoagland nutrient solution (pH = 5.8–6.0) for 10 days. Then, the following treatments were applied: control (1/2 Hoagland nutrient solution), Xh (1/2 Hoagland nutrient solution plus 20 μg $L^{-1}$ Xh), Cd (1/2 Hoagland nutrient solution with 50 μM $CdCl_2$), and Cd+Xh (1/2 Hoagland nutrient solution with 50 μM $CdCl_2$ plus 20 μg $L^{-1}$ Xh). The nutrient solution was changed every 5 days for all treatments. Following a 30-day treatment period, measurements were taken for the plant height, root length, and biomass of different tissues. The relative chlorophyll content of the leaves (specifically, the fifth fully unfolded leaf from the top) was quantified using a SPAD-502 chlorophyll meter (Minolta Camera Co., Ltd., Tokyo, Japan). Next, the ramie seedlings were washed with 20 mM $Na_2EDTA$ and double-distilled $H_2O$ (dd$H_2O$), and the Cd attached to the surface was removed. A portion of the sample was stored at −80 °C, while another portion was subjected to drying. The growth conditions of ramie cuttings included a light/dark cycle of 14 h of light and 10 h of darkness, a light intensity of 20,000 lux, a temperature of 25 °C, and a relative humidity of 70% [9].

### 2.2. Determination of Carotenoid and Chlorophyll Contents

The levels of chlorophyll and carotenoids were determined based on the methods of Wei et al. [19]. Ramie leaves with thicker veins removed were fully ground and homogenized with precooled 80% acetone. Then, the homogenate was centrifuged at $12,000\times g$ for 30 min at 4 °C. The supernatant was collected at a predetermined volume, and the OD value was determined at 663 nm, 646 nm, and 470 nm. Finally, the contents of chlorophyll a (Chla), chlorophyll b (Chlb), and carotenoids were estimated by the formulae derived by Wintermans and De-Mots [20].

### 2.3. Determination of Root Activity

Root activity was quantified using the method of Su et al. [8], with slight modifications. Ramie root was cut into 1 mm segments and soaked in 60 mM phosphate buffer solution (PBS) (pH = 7.0). After being incubated in the dark at 37 °C for a duration of 2 h, the reaction was terminated by adding 1 M sulfuric acid. The root system was fully ground in ethyl acetate, and a fixed volume was obtained. The OD value was determined at 485 nm.

### 2.4. Determination of $H_2O_2$, MDA, and Proline Contents

The $H_2O_2$ content was quantified using the method of Gong et al. [6], with slight modifications. Briefly, the tissue was ground with liquid nitrogen, and subsequently homogenized with pre-cooled 0.1% (*w/w*) trichloroacetic acid (TCA). The resulting mixture was then centrifuged at $12,000\times g$ for 10 min at 4 °C. PBS (pH = 7.0) and 1 mM KI were added to the supernatant, and the OD value was determined at 390 nm.

The malondialdehyde (MDA) content was quantified using the method of Velikova et al. [21]. After grinding ramie tissue with liquid nitrogen, 10% (*w/w*) TCA was added, fully homogenized, and centrifuged at $12,000\times g$ for 10 min at 4 °C. The supernatant was mixed with 10% TCA containing 0.5% TBA, heated at 95 °C for 30 min, cooled in an ice bath, and centrifuged at $12,000\times g$ for 10 min. The OD value of the supernatant was determined at 532 nm and 600 nm (non-specific absorption value).

The proline content was quantified using the method of Su et al. [7]. Briefly, the sample was pulverized, homogenized with 3% sulfosalicylic acid homogenate, and subsequently subjected to centrifugation at $5000\times g$ for 5 min. The supernatant was mixed with an acid ninhydrin solution and glacial acetic acid, mixed well, and then heated in a boiling water bath for 1 h before being cooled on ice water. Toluene was mixed, and the OD value was determined at 520 nm.

### 2.5. Determination of the Activity of Antioxidant Enzymes

The ramie sample was oroughly crushed in an ice water bath with PBS, followed by centrifugation at $10,000\times g$ for 15 min at 4 °C. The supernatant was the crude extract of antioxidant enzymes. Superoxide dismutase (SOD) activity was determined by Beauchamp and Fridovich [22]. Briefly, the crude enzyme solution was mixed with PBS, L-methionine (Met), nitro blue tetrazolium salt (NBT), thylene diamine tetraacetic acid (EDTA-$Na_2$), and riboflavi. The mixture was thoroughly combined and incubated under a fluorescent lamp for 15 min, and the OD value was determined at 485 nm. Catalase (CAT) activity was determined by Pine et al. [23]. Briefly, the crude enzyme extract was mixed with PBS and $H_2O_2$, mixed well, and then reacted at 25 for 2 min, and the OD value was determined at 240 nm. Ascorbate peroxidase (APX) activity was determined by Parida et al. [24]. Briefly, the crude enzyme extract was mixed with PBS, EDTA-$Na_2$, ascorbic acid, and $H_2O_2$, mixed well, and then reacted at 25 °C for 1 min, and the OD value was determined at 290 nm. Peroxidase (POD) activity was determined by Pine et al. [23]. Briefly, the crude enzyme extract was mixed with o-methoxy-phenol and $H_2O_2$, mixed well, and then reacted at 25 °C for 10 min. Metaphosphoric acid was added to terminate the reaction, and the OD value was determined at 470 nm.

### 2.6. Subcellular Component Separation

Following the methodology outlined by Wu et al. [25], cells were divided into cell wall, vacuole, organellar, and soluble fractions. The plant sample was ground with liquid nitrogen, and the pre-cooled solution containing 250 mM sucrose, 50 mM Tris-HCI (pH = 7.5), and 1.0 mM DTT was added and fully homogenized. The homogenate was passed through an 80 μm nylon cloth, and the filtered residue was obtained as the cell wall fraction. The filtrate was centrifuged at $1500\times g$ and 4 °C for 15 min, resulting in the precipitation of the vacuolar fraction. Further centrifugation of the supernatant solution at 12,000 $g$ for 30 min led to the formation of the cell organelle fraction as the precipitate, while the remaining supernatant constituted the soluble solution.

### 2.7. Isolation of the Cell Wall and Extraction of Hemicellulose

The cell wall separation and hemicellulose extraction followed the method described by Ma et al. [11]. The specific steps were as follows: First, the ramie sample was pulverized by liquid nitrogen and then immersed in an ice water bath containing 75% ice-cold ethanol for 20 min. Subsequently, centrifugation was performed at 8000 rpm for 10 min at 4 °C. The precipitates were washed and centrifuged with acetone, methanol/chloroform, and methanol. The precipitates were collected as the cell wall fraction. The cell wall was added to $ddH_2O$, heated for 1 h in a boiling water bath, and then centrifuged at $13,400\times g$ for 15 min. The precipitate was incubated with 4% NaOH and 24% NaOH for 24 h, and the supernatant was centrifuged at 16,800 rpm for 10 min at 4 °C to obtain hemicellulose.

### 2.8. The Content of Cd Absorbed by Root Cell Wall

The methods of Shi et al. [26] were used to evaluate Cd adsorption, with slight modifications. The cell wall was added to the solution (0.5 mM $CaCl_2$ solution containing 50 μM $CdCl_2$ (pH = 5.6)), which was placed in a shaker at 150 $g$ for 36 h. Subsequently, the sample was centrifuged and washed repeatedly with 0.5 mM $CaCl_2$ solution and $ddH_2O$.

### 2.9. Determination of the Hemicellulose Content

The hemicellulose content was quantified using the method of Ma et al. [11]. In particular, a volume of 500 μL of the hemicellulose extract was combined with 2.5 mL of 98% $H_2SO_4$ and 25 μL of 80% phenol and incubated for 15 min at 25 °C, followed by heating for 15 min in a boiling water bath. Subsequently, the mixtures were cooled to room temperature, and the OD value at 490 nm was measured.

### 2.10. Determination of the Cd Content

The sample was digested by $HNO_3$:$HClO_4$ (3:1, $v/v$) at a temperature of 180 °C until the liquid color became clear without precipitation, and the Cd content was measured by a flame atomic absorption spectrometer.

### 2.11. Statistical Analysis

The data are presented as the mean $\pm$ standard deviation (SD). To compare the treatments, a one-way analysis of variance (ANOVA) was conducted, followed by Tukey's tests with a significance level of $p < 0.05$. The statistical analysis was performed using SAS 9.4 (SAS Institute, Cary, NC, USA).

## 3. Results

### 3.1. Effect of Xh on Ramie Growth

To explore the effect of Xh on the growth of Ramie, we compared the phenotypes of ramie seedlings under different treatments. Ramie growth was significantly lower under Cd stress than without Cd treatment (Figure 1a). In the presence of Cd stress, the external application of Xh resulted in a significant enhancement of both plant height and root length in ramie seedlings (Figure 1a). The addition of Xh led to a 23.16% increase in plant height and a 19.15% increase in root length, compared to estimates without Xh (Figure 1b,c).

Additionally, the addition of Xh reduced the inhibitory effect of Cd stress on the biomass of ramie (Figure 1d), and the fresh weights of roots, stems, and leaves increased by 86.07%, 92.45%, and 157.58%, respectively. In short, Cd treatment significantly inhibited the main root length, plant height, and biological yield, while the external application of 20 μg/L Xh significantly reduced the toxicity of Cd.

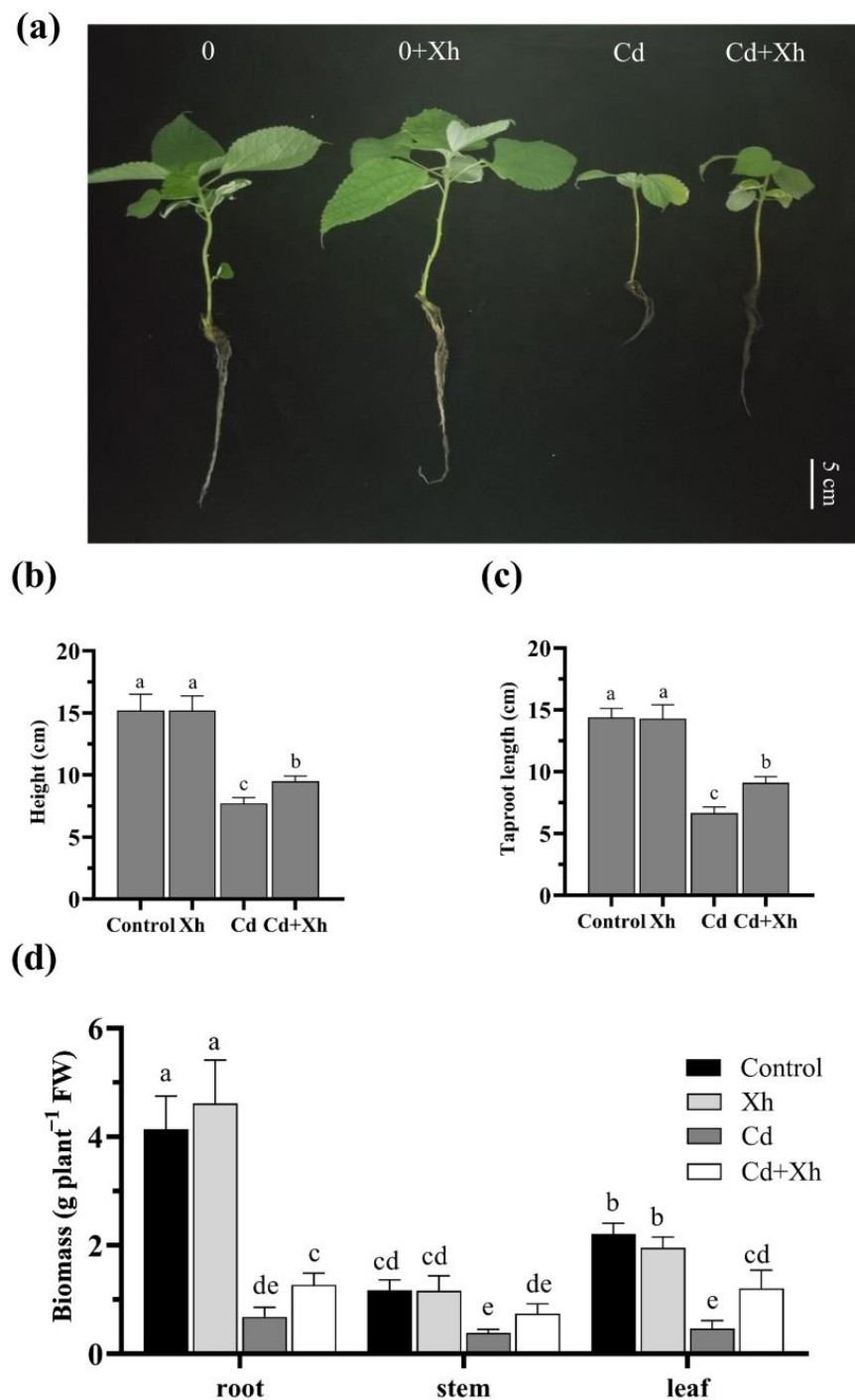

**Figure 1.** Effect of Xh and/or Cd treatment on the taproot length, plant height, and biomass of ramie: (**a**) growth performance of ramie; (**b**) plant height; (**c**) taproot length; (**d**) fresh weight of the root, stem, and leaf of ramie. Data are presented as mean ± SD (n = 9). Significant differences at $p < 0.05$ are indicated by different letters.

### 3.2. Xh Regulates Leaf Pigment and Root Activity

The SPAD value, chlorophyll content, and carotene content of leaves were further measured (Figure 2a–c). The SPAD value, Chla content, Chlb content, and carotenoid content of leaves were significantly lower under Cd treatment than without Cd ($p < 0.05$). The external application of Xh significantly attenuated the damage to photosynthetic pigment caused by Cd and significantly increased the SPAD value, Chla content, Chlb content, and carotene content of leaves. Additionally, Cd stress significantly decreased root activity ($p < 0.05$), and root activity was significantly increased at Xh treatment ($p < 0.05$; Figure 2d).

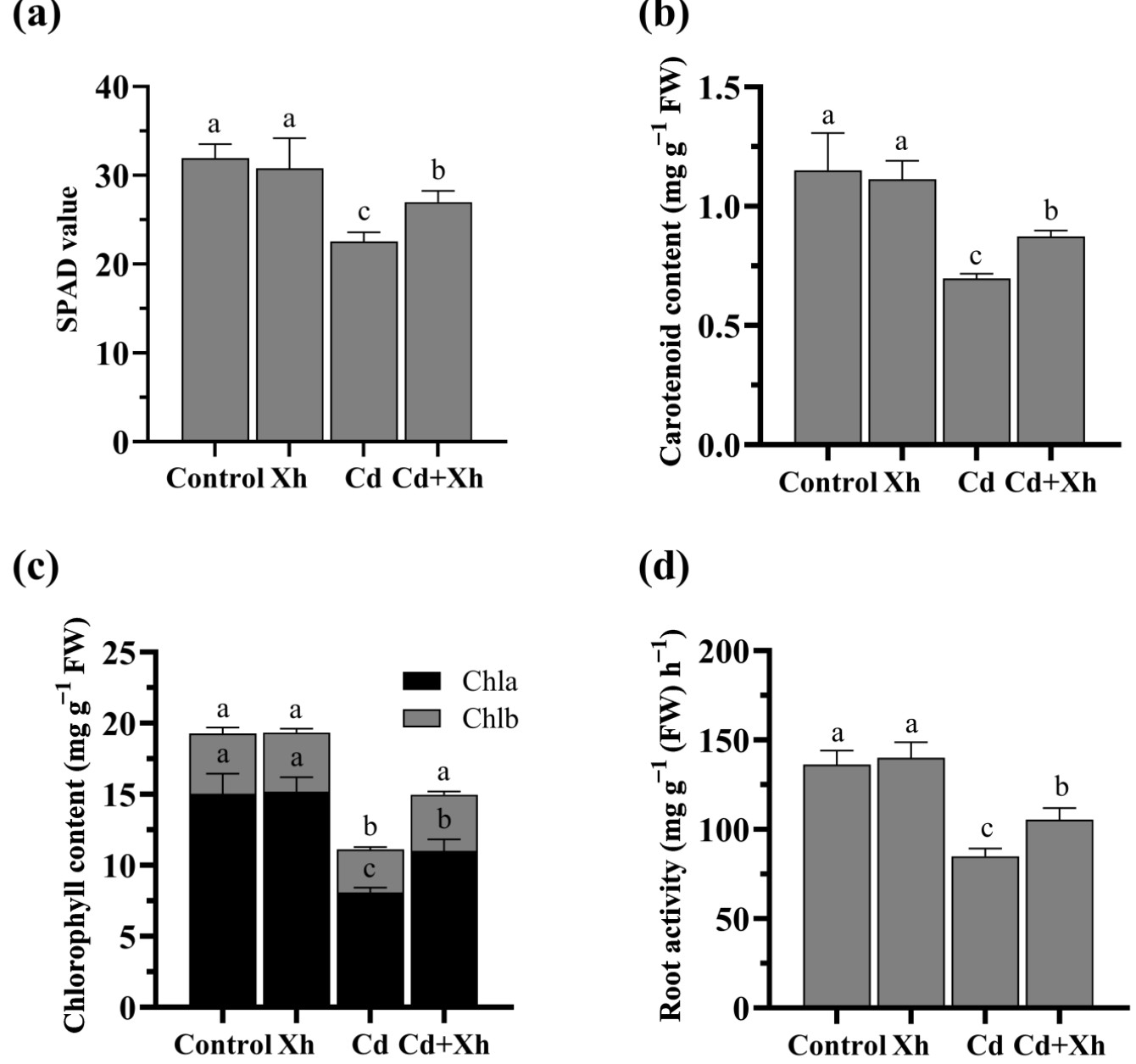

**Figure 2.** Effect of Xh and/or Cd treatment on (**a**) SPAD content, (**b**) carotenoid content, (**c**) contents of Chla and Chlb, and (**d**) root activity of ramie. Data are presented as mean ± SD (n = 3). Significant differences at $p < 0.05$ are indicated by different letters.

### 3.3. Xh Reduces Oxidative Damage and Improves the Activity of Antioxidant Enzymes

To determine whether the external application of Xh can improve the Cd tolerance of plants, $H_2O_2$, MDA, and proline levels in ramie were evaluated (Figure 3). No significant differences were observed in the levels of $H_2O_2$, MDA, and proline between the control and Xh treatments. However, under Cd and Cd+Xh treatments, the levels of $H_2O_2$, MDA, and proline in ramie were significantly higher than those in the control ($p < 0.05$). The levels of $H_2O_2$, MDA, and proline in the root and shoot were significantly reduced by Xh treatment under Cd stress ($p < 0.05$). In short, the external application of Xh can regulate ROS scavenging, alleviate membrane lipid peroxidation caused by Cd, and reduce Cd toxicity in ramie.

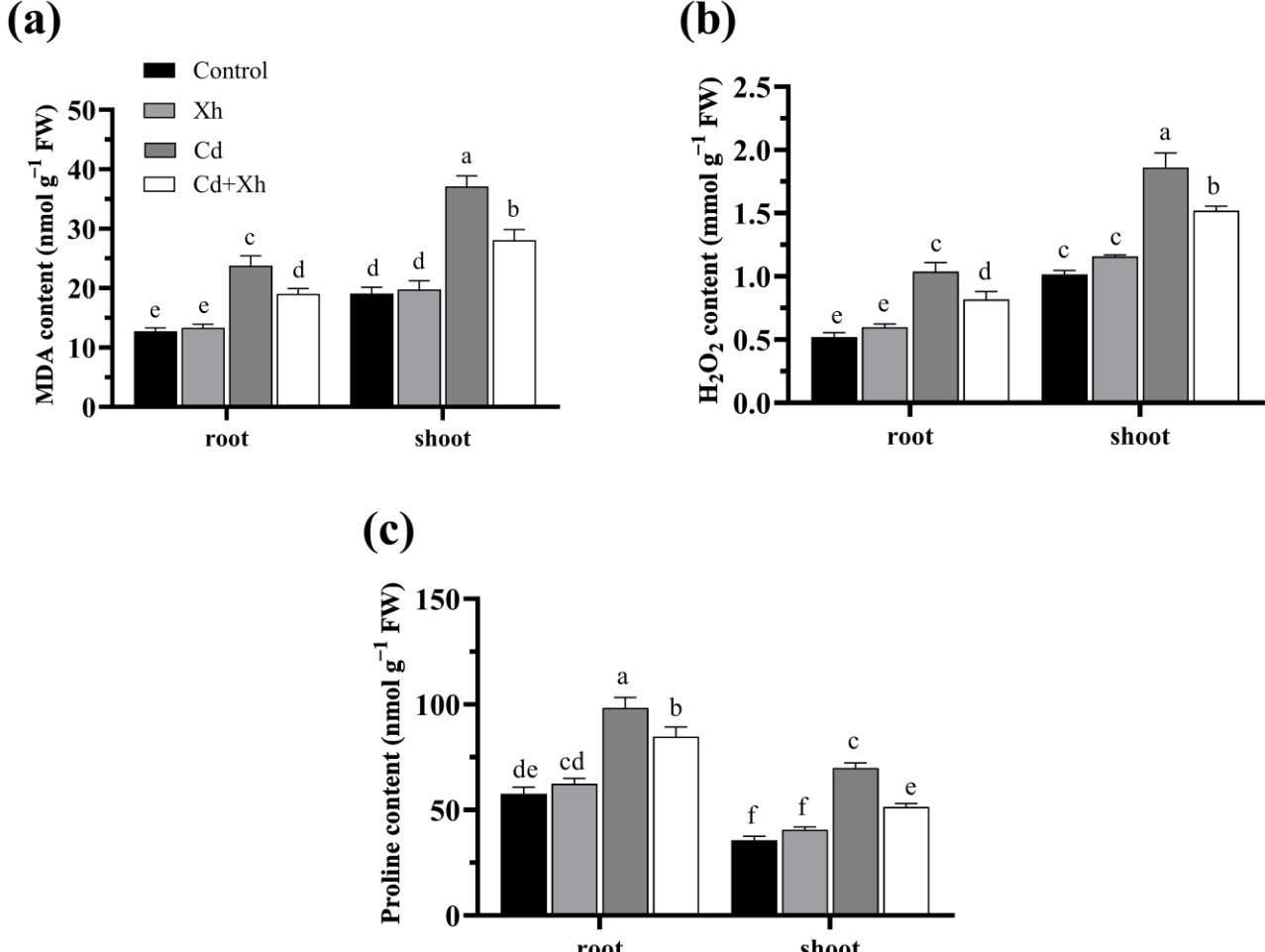

**Figure 3.** Effect of Xh and/or Cd treatment on (**a**) MDA content, (**b**) $H_2O_2$ content, and (**c**) proline content. Data are presented as mean $\pm$ SD (n = 3). Significant differences at $p < 0.05$ are indicated by different letters.

We also evaluated the activities of SOD, CAT, APX, and POD in ramie (Figure 4). There were no significant differences in the activities of SOD, CAT, APX, and POD between the shoot and root of ramie treated with Xh alone ($p > 0.05$). The activities of SOD, CAT, APX, and POD in the shoot and root of ramie were significantly lower after treatment with Cd alone than in the control ($p < 0.05$). The external application of Xh significantly increased the activities of SOD, CAT, APX, and POD in the shoot and root of ramie under Cd stress ($p < 0.05$).

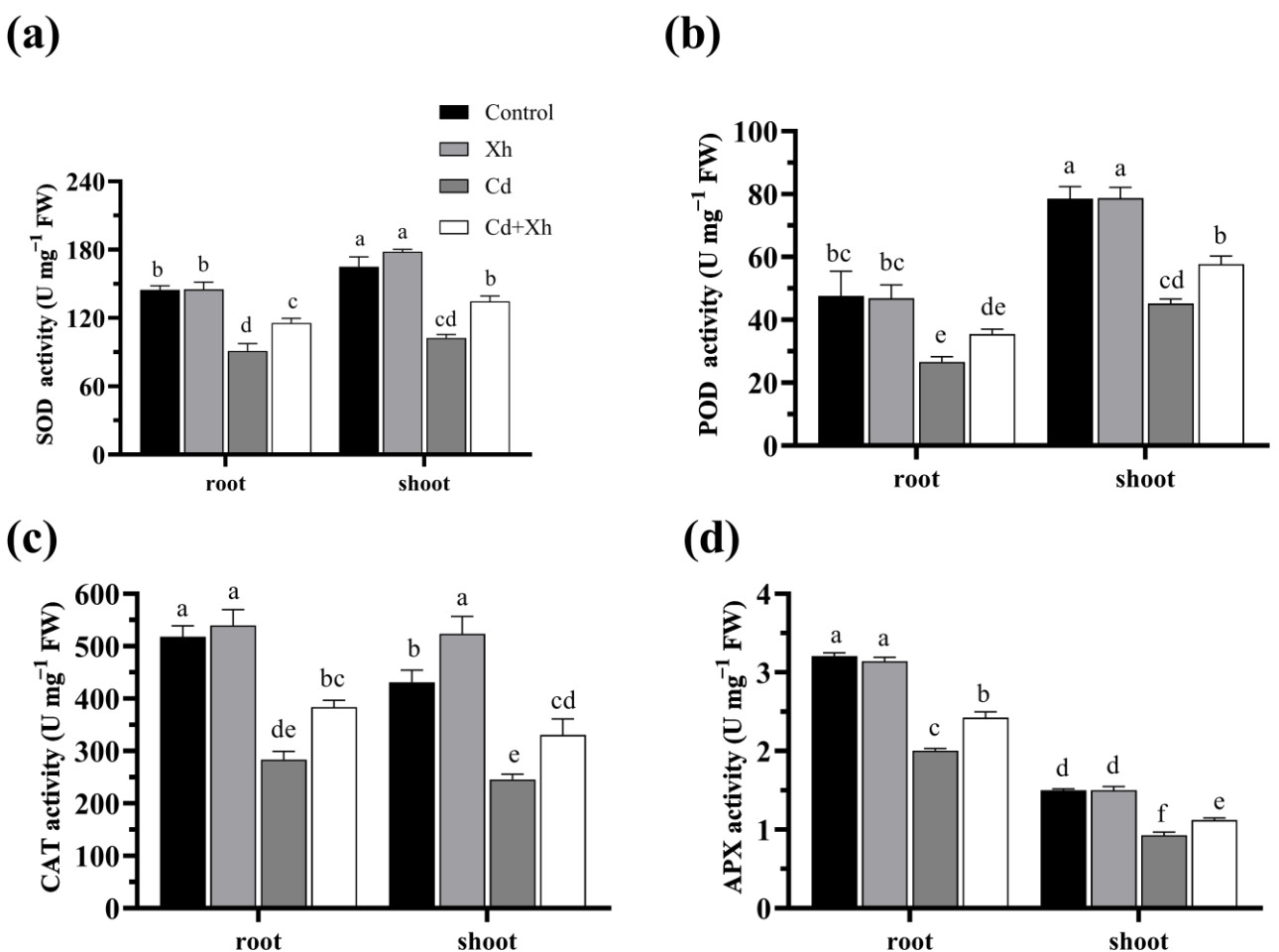

**Figure 4.** Effect of Xh and/or Cd treatment on the activities of (**a**) SOD, (**b**) POD, (**c**) CAT, and (**d**) APX in ramie. Data are presented as mean $\pm$ SD (n = 3). Significant differences at $p < 0.05$ are indicated by different letters.

*3.4. Xh Regulates the Distribution of Cd in Cells*

We further analyzed the Cd contents in the shoot and root of the ramie. In comparison to the single Cd treatment, in the Cd+Xh group, we observed a significant increase in the Cd content in the root ($p < 0.05$; Figure 5a), while there was a significant decrease in the Cd content in the shoot ($p < 0.05$; Figure 5b). These results show that Xh can increase Cd fixation in the root and reduce transfer to the shoot, thereby reducing Cd toxicity.

Furthermore, we evaluated the subcellular distribution of Cd in the root and shoot of ramie. Overall, we found that Cd was mainly enriched in the cell wall (60.05–68.36%), while the content in the soluble fraction was lower (2.76–4.02% of the total Cd content). Under Cd stress, supplementation with Xh increased the proportion of bound-Cd in the shoot cell wall from 60.05% to 64.99%. Simultaneously, the proportion of Cd in vacuoles increased from 16.32% to 19.01%, the proportion of Cd in the soluble fraction decreased from 4.02% to 3.80%, and the proportion of Cd distributed in organelles decreased from 19.61% to 12.20% (Figure 5c,d). Furthermore, the content and proportion of Cd in the cell wall and vacuole of the ramie root increased significantly in the Cd+Xh, while the proportion of Cd in organelles decreased significantly ($p < 0.05$; Figure 5e,f). This indicated that Xh modifies the subcellular distribution of Cd in ramie cells.

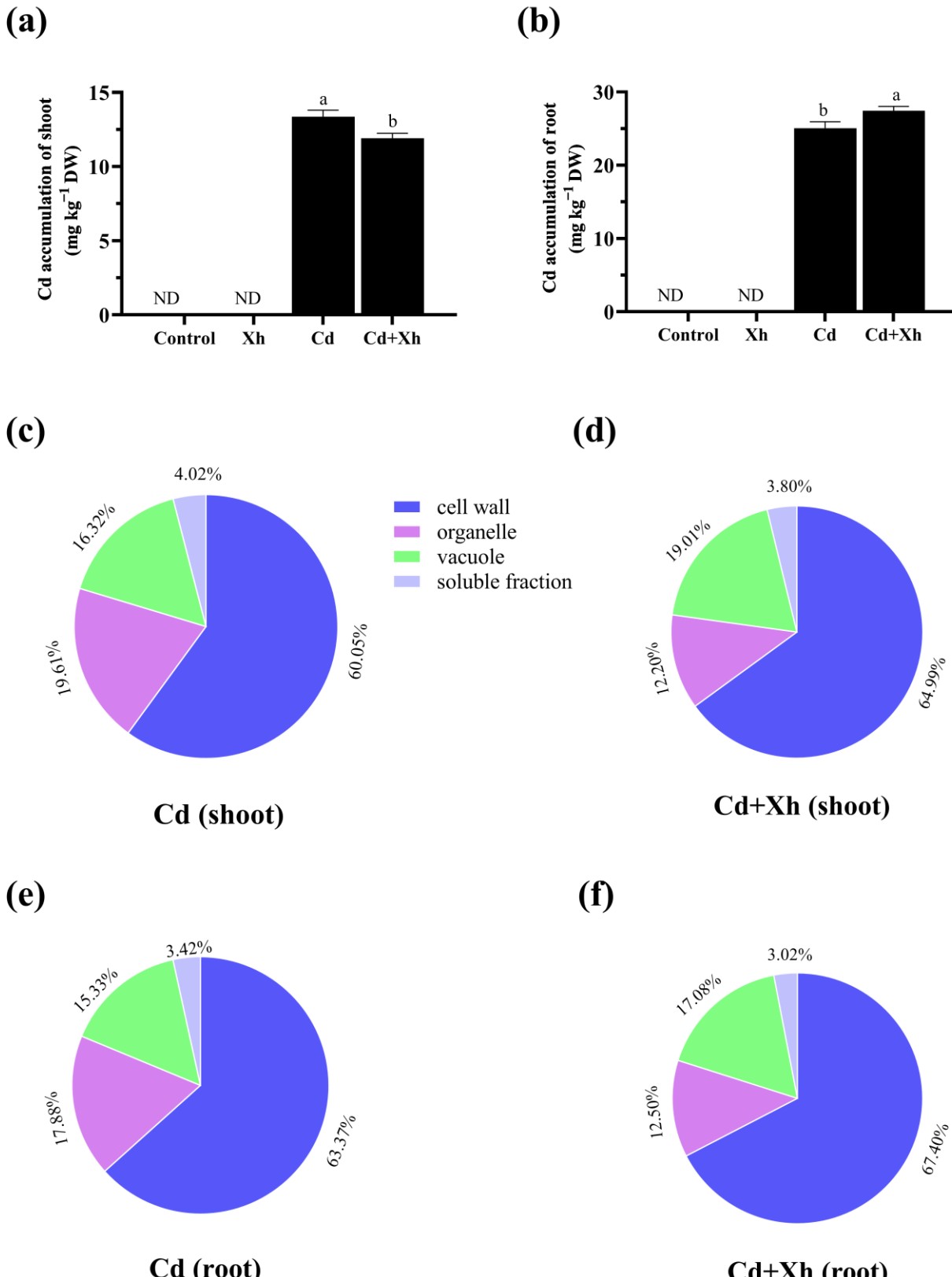

**Figure 5.** Effect of Xh and/or Cd treatment on Cd distribution in different tissues and cells of ramie. (**a**,**b**) the Cd contents of the root and shoot of ramie, respectively. (**c**,**e**) the subcellular fractions of Cd in the root and shoot of ramie under Cd stress, respectively. (**d**,**f**) are the subcellular fractions of Cd in the root and shoot of ramie under Cd stress after treatment with Xh, respectively. Data are presented as mean ± SD (n = 3). Significant differences at $p < 0.05$ are indicated by different letters.

### 3.5. Xh Regulates the Hemicellulose Content and Cd in Hemicellulose

Cd accumulation in the cell wall of root tissues was significantly higher under Cd+Xh treatment compared to Cd treatment alone ($p < 0.05$; Figure 6a). Similarly, the absorption of Cd by the root cell wall was significantly enhanced under the Xh treatment compared to the control ($p < 0.05$; Figure 6b), indicating that the addition of Xh promotes Cd fixation on the cell wall.

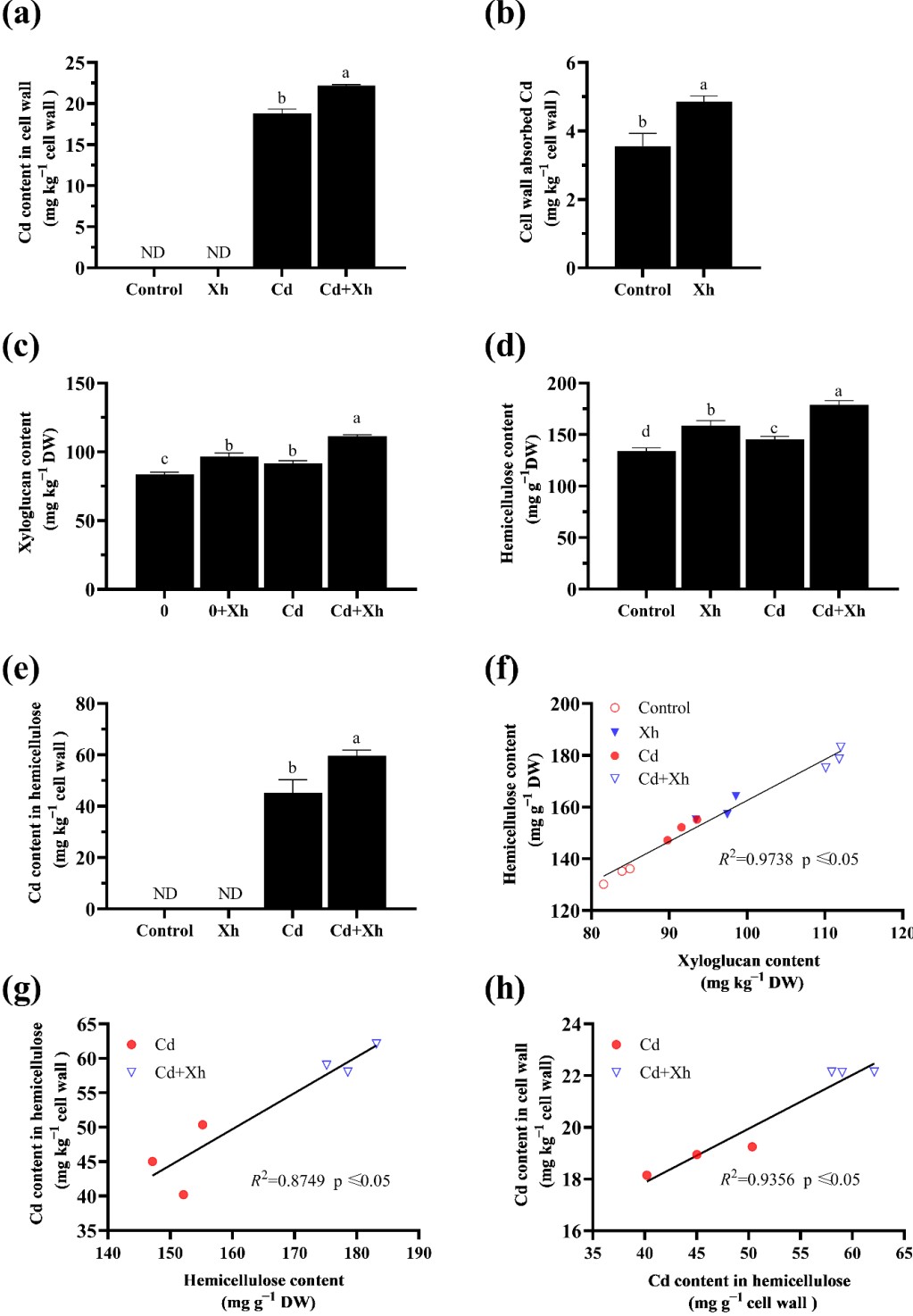

**Figure 6.** Effect of Xh and/or Cd treatment on the Cd content, hemicellulose content, and Cd content on hemicellulose in the ramie cell wall. (**a**) Cd concentration in ramie root cell walls; (**b**) Adsorption

capacity of ramie root cell walls grown for 30 days in the presence or absence of xyloglucan incubated with 50 μM CdCl$_2$ for 36 h. (**c–e**), xyloglucan content, hemicellulose content, and Cd content in hemicellulose of ramie roots under Cd stress for 30 days under control, Xh, Cd, and Cd+Xh treatments, respectively. (**f**) Correlation between the xyloglucan content and hemicellulose content per ramie plant with and without Cd stress for 30 days in the presence or absence of Xh. (**g**) Correlation between the hemicellulose content and Cd content in hemicellulose per ramie plant under Cd stress for 30 days in the presence or absence of Xh; (**h**) Correlation between the Cd content in hemicellulose and cell wall of ramie per plant under Cd stress for 30 days in the presence or absence of Xh. Data are presented as mean $\pm$ SD (n = 3). Significant differences at $p < 0.05$ are indicated by different letters.

We have previously found that hemicellulose is a major component responsible for Cd binding in ramie, and variation in rates of accumulation in ramie germplasm is related to the hemicellulose content [11]. Because xyloglucan is an important component of hemicellulose, we determined the hemicellulose content and Cd of hemicellulose in ramie roots. The hemicellulose content was significantly higher in the Xh treatment than in the 0 group ($p < 0.05$) and was further increased by the combination of Cd+Xh ($p < 0.05$; Figure 6c). Additionally, we found that Cd accumulation via hemicellulose in ramie was significantly higher under Cd+Xh treatment compared to Cd treatment alone with the addition of Xh compared to Cd treatment alone (Figure 6d).

We further evaluated the relationships among the xyloglucan content, hemicellulose content, Cd content on hemicellulose, and Cd content on the cell wall. A strong and statistically significant linear correlation was observed between the xyloglucan content and hemicellulose content, with a fitting curve of $y = 1.5879x + 3.7507$ ($R^2 = 0.9738$) (Figure 6f). There was also a strong and statistically significant linear correlation between the Cd content in hemicellulose and hemicellulose content, and the fitting curve was $y = 0.2083x + 9.5309$ ($R^2 = 0.9356$) (Figure 6g). There was a highly significant linear correlation between the Cd content in hemicellulose and the Cd content in the cell wall, and the fitting curve was $y = 0.5253x - 34.337$ ($R^2 = 0.8749$) (Figure 6h). These findings indicated that increasing Xh could increase the hemicellulose content and thereby increase the Cd content in the cell wall.

## 4. Discussion

Cd can weaken plant photosynthesis and disrupt the water balance, leading to plant cell damage and ultimately affecting the growth and development of plants [27,28]. In this study, the root growth, plant height, and biological yield of ramie were inhibited under Cd stress (Figure 1). This inhibition of plant growth by Cd has been observed in other plants, such as tobacco [7], peanut [29], and cucumber [30]. The addition of Xh attenuated the inhibitory effect of Cd in ramie, indicating that Xh could improve Cd tolerance. Cd-induced increases in ROS production can destroy the lipid membrane of chloroplasts or increase oxidation, affecting the chlorophyll content and inhibiting photosynthesis [31]. In this study, Cd stress significantly decreased the SPAD value and photosynthetic pigment content of ramie leaves, and the external application of Xh could alleviate these Cd-induced changes (Figure 2a–c). Carotenoids are not only photosynthetic pigments but also endogenous antioxidants with an important role in ROS scavenging [32,33]. We observed that the Cd-induced decreases in MDA, H$_2$O$_2$, and proline levels in ramie were attenuated by the addition of Xh (Figure 3). These results showed that the addition of Xh enhanced the ROS scavenging ability in ramie leaves, alleviated chlorophyll damage, and inhibited the degradation of photosynthetic pigments. In addition, we measured root activity in ramie as an indicator of the ability to absorb water and nutrients, which is a crucial factor in facilitating plant growth [34]. In this study, Cd significantly reduced root activity in ramie, and this Cd-induced decrease was significantly attenuated by the application of Xh (Figure 2d). The root growth of ramie treated with Xh was significantly greater than that of ramie treated with Cd alone (Figure 1b). This suggests that Xh may increase root activity,

thus promoting plant growth under Cd stress. These findings further proved that Xh can improve the Cd tolerance of ramie.

MDA has a peroxidation effect on cellular lipids, reflecting the degree of plant cell damage [35]. Similarly, the content of $H_2O_2$ and proline indicates cellular lipid peroxidation and can reflect the degree of plant cell membrane damage [6,36,37]. The induction of Cd stress can elicit a surge in ROS levels within plant cells, and ROS can attack polyunsaturated fatty acids, causing lipid peroxidation and oxidative damage to plants [38,39]. In this study, the levels of MDA, $H_2O_2$, and proline in ramie increased significantly under Cd stress (Figure 3), consistent with previous research results [40]. However, after adding Xh, the levels of MDA, $H_2O_2$, and proline exhibited a significant reduction compared to those under single Cd stress (Figure 3). It is possible that the application of Xh reduced the oxidative damage caused by Cd. To reduce intracellular oxidative damage and maintain intracellular homeostasis, plants regulate the expression of genes associated with antioxidant defense mechanisms. [41,42]. POD, SOD, CAT, and APX are four crucial antioxidant enzymes [43]. SOD is responsible for the conversion of $O_2^-$ to $H_2O_2$, which is the first step to removing ROS. POD, CAT, and APX can convert $H_2O_2$ into $O_2$ and $H_2O$, thus reducing the toxicity of ROS [44]. The results showed that compared with plants exposed solely to Cd stress, ramie treated with Xh exhibited significantly elevated activities of SOD, CAT, APX, and POD in the root and shoot (Figure 4). This enhancement in antioxidant enzyme activity increased the $H_2O_2$ and ROS scavenging ability and plant tolerance to Cd [40].

Plants have two general strategies to resist heavy metal stress: internal detoxification mechanisms and the prevention of metal ions from entering cells [45]. Internal detoxification mainly involves the transport of metal ions to vacuoles in the form of metal-organic acid chelates [46] or the chelation of heavy metals to reduce toxicity [47]. In this study, Cd accumulation in the root of the Cd+Xh treatment group increased, while that in the shoot was relatively low, indicating a decrease in the mobility of Cd (Figure 5a,b). This finding is consistent with previous research by Wang et al. [48], who demonstrated that selenium can alleviate toxicity by reducing Cd mobility in ramie. It is possible that the root has stronger Cd fixation ability than that of the shoot and inhibits transport to the shoot. As the first barrier, the cell wall prevents Cd from entering the cells and serves as a major site for metal ion fixation [11,49]. Our study revealed that the cell wall was indeed a significant site for Cd accumulation in ramie cells. Xh regulated the intracellular distribution of Cd and significantly increased Cd accumulation in the cell wall (Figure 5c–f). Additional in vitro assays also showed that Xh could substantially enhance Cd accumulation in the cell wall (Figure 6b). It should be emphasized that Cd immobilized on the cell wall can be converted into the PC-Cd complex and then transferred to vacuoles [50], which may explain our observation of increased Cd content in vacuoles in the Cd+Xh treatment. However, further molecular and physiological analyses are needed to support this hypothesis. The cell wall comprises pectin, cellulose, hemicellulose, and lignin [51]. Among these components, hemicellulose has been recognized as the major target of Cd in plant cells [52]. Similarly, alterations made either to the composition or constituents comprising the cell wall can influence its affinity for metal ions [53,54]. In this study, compared with those for Cd treatment, Cd+Xh treatment resulted in a significantly higher concentration of hemicellulose and Cd in hemicellulose (Figure 6c,d). Furthermore, we detected highly significant linear correlations between the Cd content in the cell wall and the Cd content in hemicellulose (Figure 6h) and between the Cd content in hemicellulose and the hemicellulose content (Figure 6g). These results show that Xh can effectively increase the hemicellulose content while acilitating Cd accumulation in the cell wall. As early as 1993, scholars have proposed that the oligosaccharides that make up the cell wall may act as signaling substances for the growth and development processes of plants [14]. Exogenous Xh is transferred and integrated into plants to form a longer xyloglucan chain via xylanase, which in turn affects the xyloglucan content in plants [55]. Therefore, we speculate that the external application of Xh can induce hemicellulose synthesis in the plant cell wall, thereby enhancing the

Cd binding ability of the cell wall and improving the Cd tolerance of ramie. However, additional molecular and genetic studies are needed to evaluate this idea.

## 5. Conclusions

To sum up, the exogenous application of Xh could enhance the Cd fability of the cell wall by increasing the hemicellulose content, thereby altering the subcellular distribution of Cd, reducing the intracellular Cd content, and reducing the transport of Cd to the shoot. Additionally, the exogenous application of Xh promotes ROS scavenging by up-regulating antioxidant enzyme activities to reduce membrane lipid peroxidation. Ultimately, these mechanisms mitigate the inhibitory effects of Cd on ramie growth and development (Figure 7). In summary, this study provided a novel perspective on enhancing the adaptability and soil remediation ability of ramie, as well as a new theoretical basis and direction for ramie breeding with high Cd tolerance.

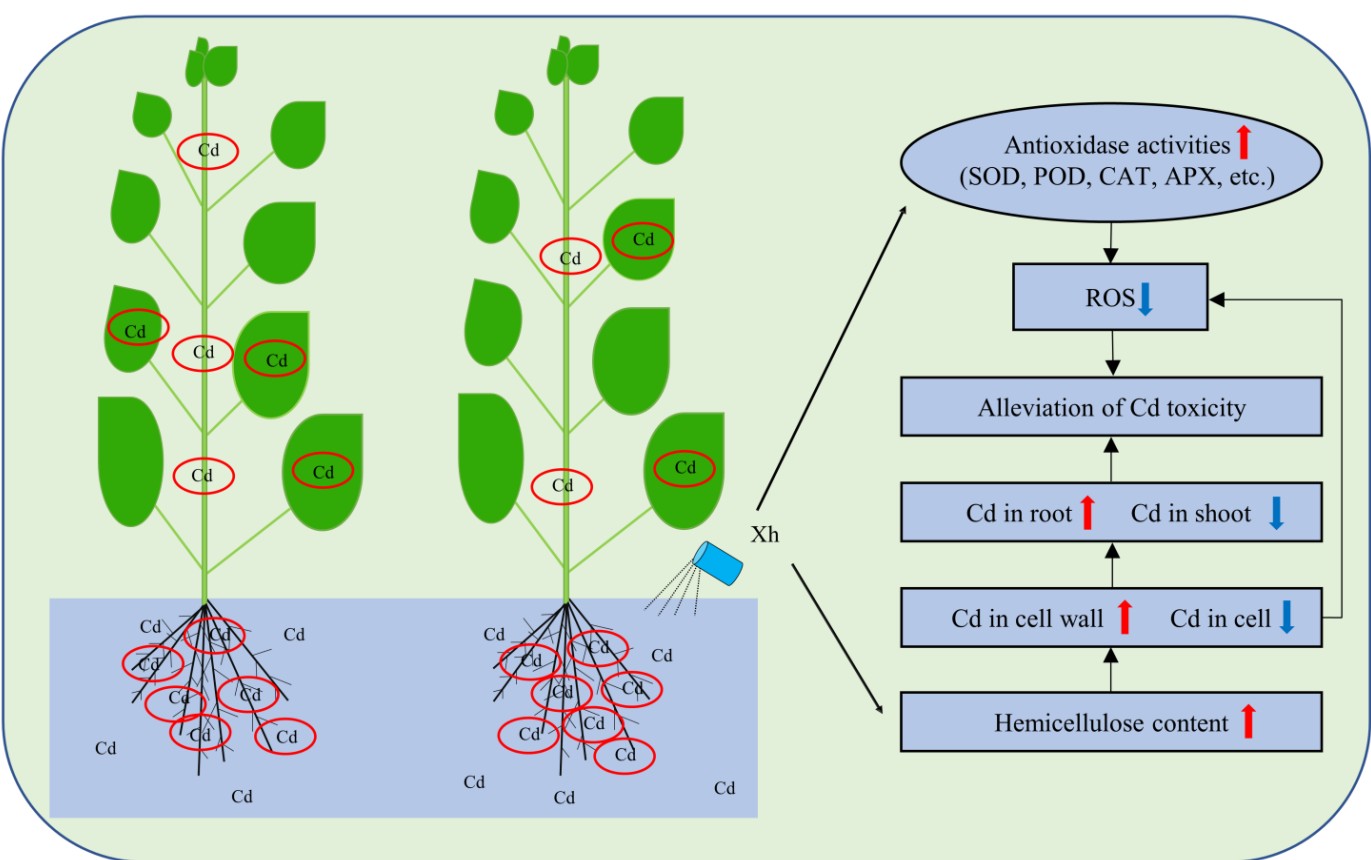

**Figure 7.** Schematic diagram of the mechanism by which exogenous Xh alleviates Cd toxicity in *Boehmeria nivea*.

**Author Contributions:** Y.M.: methodology, investigation, formal analysis, writing—original draft preparation; H.J., L.Z., Y.Z., P.H., X.L. (Xueying Lv), X.L. (Xiaochun Liu), Y.X. and Y.J.: formal analysis and visualization; Y.J.: conceptualization, supervision, and project administration. All authors have read and agreed to the published version of the manuscript.

**Funding:** This research was supported by the National Natural Science Foundation of China (31872877 and 32071940), the National Key R&D Program of China (2019YFD1002205-3), and the Research and Development Projects in the Key Area of Hunan Province (2019NK206102 and 2020NK2028).

**Data Availability Statement:** Data are contained within the article.

**Acknowledgments:** We would like to thank Weidan Yin, Huangqiao Xiao, and Yuling Shi for their help.

**Conflicts of Interest:** The authors declare no conflict of interest.

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
