# Peer review of "Exogenous Xyloglucan Oligosaccharides Alleviate Cadmium Toxicity in Boehmeria nivea by Increasing the Cadmium Fixation Capacity of Cell Walls"

_agronomy, doi:10.3390/agronomy13112786_

Round 1

Reviewer 1 Report

Comments and Suggestions for Authors

The manuscript entitled “Exogenous xyloglucan oligosaccharides alleviate cadmium toxicity in Boehmeria nivea by increasing the cadmium fixation capacity of cell walls” describes the effects of xyloglucan oligosaccharides (Xh) on ramie (Boehmeria nivea) growth and physiology in the presence of Cd. The Authors showed that exogenous Xh can improve plant tolerance to Cd by increasing antioxidant enzyme (SOD, POD, CAT, APX) activity, reducing lipid peroxidation, and increasing Cd retention in roots and cell walls. The manuscript is interesting; however, this requires improvement. I have listed the following comments:

·        The Abstract must be improved. Some sentences are incorrect or unclear and should be changed: L13-14 (“xyloglucan oligosaccharides (Xh) are metabolized by xyloglucan,…”) and L21 (“and membrane damage due to membrane in plants…”)

·        Introduction: L61-64   the sentence is very long and, therefore, incomprehensible to the reader. I advise to rewrite it.

Material and Methods

·        It is not stated on what basis the used Cd and Xh concentrations were selected

·        L127: The references for Su et al. are [8] not [7]. Please check the reference numbers in the whole manuscript carefully.

·        L127-129: Please read the sentence and rewrite

·        L155-156: Please specify what high-speed centrifugation is and at how many g or rpm?

·        Please specify the determination of Cd content in greater detail

Results – The Figures need to be improved.

·        Figure 1a is not described in the Results section.

·        I suggest changing the construction in Figure 1d. In my opinion, it would be clearer to compare all treatments between plant organs but not organs in one treatment. In this way of presentation, the results in Figure 1d are difficult to analyze. Please change the series of figures accordingly.

·        In the Figure legends please add “effect of xyloglucan and/or Cd treatment…..

·        Units on Figure 2d are incorrect

·        Figure 3 and 4 – the same comment as to the Figure 1d, please change series in the Figures

·        I find the mistake in the L273, please check and correct the description (R2) – Figure 6f

Discussion

·        L316-318: In my opinion, this is speculative. It is necessary to determine the uptake of one of the minerals or minerals content in the tissues and, for example, the relative water content. Root activity is a general parameter. This may indicate ongoing metabolic processes in root cells, such as respiration, energy transformation, and oxidation-reduction transformations. This parameter can be defined as root vitality.

·        L351-353 Why “in vitro assay”? What is the difference between that experiment (fig6b) and the others. It does not explain enough.

Author Response

Dear Reviewer:

Thank you very much for your attention and the referee's evaluation and comments on our paper agronomy-2638757. We have revised the manuscript according to your detailed suggestions. Enclosed please find the attachment.

Thank you very much for all your help and looking forward to hearing from you soon.

Sincerely yours,

Yushen Ma

Reviewer 2 Report

Comments and Suggestions for Authors

Current research article on "Exogenous xyloglucan oligosaccharides alleviate cadmium toxicity in Boehmeria nivea by increasing the cadmium fixation capacity of cell walls" is quite interesting and prsentation of outcome are well described. I have one suggestion that germplasn describe the large number of genotypes but here only one gentype used. therefore i suggest to replace the germplasm with genotype. 

Author Response

(The authors gave the same response as above.)

Reviewer 3 Report

Comments and Suggestions for Authors

Comments on the Quality of English Language

Minor editing

Author Response

(The authors gave the same response as above.)

Round 2

Reviewer 1 Report

Comments and Suggestions for Authors

All previous reviewer comments were taken into account and corrected by the Authors.

One minor point below:

 Point 3: It is not stated on what basis the used Cd and Xh concentrations were selected.

Response 3: Thanks for the comment. We did a preliminary experiment to find the most obvious concentrations of Cd and Xh, which we talked in introduction.

Reviewer answer:

Thank you for the explanation. Then please add the references to L:78-83 describing your previous studies in the Introduction.